# Occupational Exposure to Wood Dust and the Burden of Nasopharynx and Sinonasal Cancer in Canada

**DOI:** 10.3390/ijerph19031144

**Published:** 2022-01-20

**Authors:** Amirabbas Mofidi, Emile Tompa, Christina Kalcevich, Christopher McLeod, Martin Lebeau, Chaojie Song, Joanne Kim, Paul A. Demers

**Affiliations:** 1Institute for Work & Health, Toronto, ON M5G 1S5, Canada; etompa@iwh.on.ca (E.T.); ckalcevich@hotmail.com (C.K.); chris.mcleod@ubc.ca (C.M.); 2Department of Economics, McMaster University, Hamilton, ON L8S 4L8, Canada; 3Dalla Lana School of Public Health, University of Toronto, Toronto, ON M5T 3M7, Canada; 4School of Population and Public Health, Faculty of Medicine, University of British Columbia, Vancouver, BC V6T 1Z3, Canada; 5Institut de Recherche Robert-Sauvé en Santé et en Sécurité du Travail (IRSST), Montreal, QC H3A 3C2, Canada; Martin.Lebeau@irsst.qc.ca; 6Occupational Cancer Research Centre (OCRC), Toronto, ON M5G 1X3, Canada; chaojie.song@ontariohealth.ca (C.S.); KimJ@fellows.iarc.fr (J.K.); paul.demers@ontariohealth.ca (P.A.D.)

**Keywords:** woodworking, forestry, carpentry, incidence, healthcare costs, productivity losses, lifetime costs

## Abstract

Background: Millions of workers around the world are exposed to wood dust, as a by-product of woodworking. Nasopharynx cancers (NPCs) and sinonasal cancers (SNCs) are two cancers that can be caused by occupational exposure to wood dust, but there is little evidence regarding their burden in Canada. Objective: the aim of this study was to estimate the incidence and economic burden of newly diagnosed cases of NPC and SNC in 2011 in Canada, attributable to occupational exposures to wood dust. Methods: calculating the incidence of cancer attributable to occupational exposure involved three steps of defining relative risk, assessing the prevalence of exposure and population modelling. We estimated the lifetime costs of newly diagnosed NPC and SNC from the societal perspective. The three major cost categories that we considered were direct costs (healthcare costs, out-of-pocket costs, and informal caregiving costs), indirect costs (labour productivity/output costs, employer adjustment costs, and home production losses), and intangible costs (health-related quality of life losses). To generate an estimate of economic burden, we used secondary data from multiple sources and applied them to our computational model developed from an extensive literature review. Results: From approximately 1.3 million workers exposed to wood dust, we expected 28%, 43% and 29% were exposed to low, medium, and high levels, respectively. We estimated from 235 newly diagnosed cases of NPC and 245 newly diagnosed cases of SNC, 4.6% (11 cases) and 4.4% (11 cases) were attributed to occupational exposure to wood dust, respectively. Our estimates of the economic burden of occupational NPC and SNC were about CAD 5.4 million (CAD 496,311 per-case) and CAD 6.7 million (CAD 627,437 per-case), respectively. For NPC direct costs constituted approximately 20% of all costs, and indirect and intangible costs accounted for 55% and 25%, while for SNC the breakdown distribution were 16%, 42% and 42%, respectively. Conclusions: Our estimates highlighted the importance of occupational NPC and SNC amongst other occupational cancers, especially in countries with large wood-related industries. This paper also serves the information needs of policymakers who are seeking to make evidence-based decisions about occupational cancer prevention efforts.

## 1. Introduction

Wood dust is considered carcinogenic to humans (Group 1) according to the International Agency for Research on Cancer (IARC) [1]. Nasopharynx cancer (NPC) and sinonasal cancer (SNC) are two important cancers caused by exposure to wood dust. The incidence of NPC and SNC was estimated higher for men than women, mainly because of the historical gender composition of workforce in wood-dust-exposed occupations [2,3]. Other health issues associated with exposure to wood dust included respiratory disease such as asthma, bronchitis, chronic lung function impairment [4] and the development of allergic symptoms [5].

Millions of workers around the world are exposed to wood dust, as the by-product of woodworking. Forestry is one of the industrial sectors whose workers are at a considerable risk of exposure to wood dust [6]. Canada has a diverse forestry industry that employs about 210,000 workers across different provinces, and it accounted for about 7.2% of total exports in 2017 [7]. However, occupational exposure to wood dust is not limited to the forestry industry, and industrial sectors that use wood are frequently widespread, i.e., construction, manufacturing and services, such as carpenters, also reported high level of wood dust exposure [8,9].

Although some studies reported prevalence of exposure to wood dust in workplaces in Canada, there is not much evidence regarding the incidence of attributed occupational NPC and SNC and their economic burden to society. Workers’ compensation claims provide some insights into the magnitude of such a burden, but their approximation does not necessarily represent the whole picture. Though occupational cancers are compensable under most workers’ compensation programs, few such cases make their way into that system in Canada due to their long latency period. Furthermore, there are other costs that are not captured in workers’ compensation systems. For instance, from an employer’s perspective, costs incurred for the recruitment and training of a replaced worker or costs associated with accommodating a claimant are not captured [10]. From worker’s perspective, the loss of home production, informal care provided by family members or friends, and the intrinsic value of health are some cost components that are not usually captured in workers’ compensation systems.

In a review of studies conducted by the research team, we were unable to identify peer-reviewed economic burden studies focused solely on NPC and SNC attributed to occupational exposure in Canada. The only closely related study identified is that of Orenstein et al., who estimated the economic burden of occupational cancers in Alberta for 2003 [11]. They estimated the costs of nose and nasal sinuses cancers at CAD 15,106 per-case in 2008 Canadian dollars (CAD 16,507 in 2011 Canadian dollars), however they did not distinguish between the NPC and SNC. Their total costs were divided into direct and indirect costs. Under the direct costs, since the researchers have only access to the healthcare costs for lung cancer, they used a proportion of the lung cancer costs to estimate nose and nasal sinuses healthcare costs (a ratio of 0.60). Under the indirect costs, they considered absenteeism, short and long-term disability, and premature mortality.

In the United States (US), Jacobson et al. estimated the healthcare costs of head and neck cancers between 2004 and 2008 [12]. They estimated healthcare costs of all head and neck tumors arising from five primary sites, i.e., larynx, pharynx, oral cavity, salivary glands and paranasal sinuses. They identified the annual average healthcare costs for patients during the year after diagnosis at CAD 79,151 per-case in 2009 US dollars (CAD 95,848 per-case in 2011 Canadian dollars). They calculated the direct healthcare costs of the cases, based on the analysis of administrative claims data, considering charges for inpatient admissions, outpatient hospital visits, office visits, emergency department visits, and outpatient prescription drugs. They also estimated the losses as a result of short-term disability for one year following the index date at CAD 7952 per-case in 2009 US dollars (CAD 9629 per-case in 2011 Canadian dollars).

In another study in the U.S., Epstein et al. estimated healthcare costs associated with treating oral and pharyngeal squamous cell carcinoma for the years 1995 to 2003 [13]. They estimated the average healthcare costs for those treated within the first year following initial diagnosis at USD 25,319 per-case in 2002 US dollars (CAD 51,895 per-case in 2011 Canadian dollars). They used administrative claims data to calculate direct payments for diagnosed patients and their total medical costs included amounts paid for inpatient, outpatient, long-term care, and prescription services.

Across all studies identified, most of them exclusively focused on healthcare costs, though they did not consider out-of-pocket costs or informal caregiving costs. Furthermore, no studies considered health-related quality of life losses associated with NPC and SNC. Given the lack of knowledge regarding the burden of occupational NPC and SNC in Canada, this study was designed to fill the gap and to estimate incidence and economic burden of newly diagnosed occupational NPC and SNC attributed to exposure to wood dust. We used an incidence cost approach to estimate the economic burden of newly diagnosed cases for the year 2011 under three broad categories of direct, indirect and intangible.

## 2. Materials and Methods

To generate an estimation on the incidence and economic burden of occupational NPC and SNC, we combined primary and secondary data sources with our model assumptions. Our input data were retrieved from the Occupational Cancer Research Centre (OCRC) [14], CAREX Canada (2018) [15], Canadian population life tables (2010–2012) [16], schedule facility fees and physician services under the health insurance act of Ontario (2013) [17], Canadian Labour Force Survey (LFS) (2011) [18], Survey of Labour and Income Dynamics (SLID) (2011) [19], Canadian System of National Accounts (CSNA) (2010) [20], Survey of Employment, Payrolls and Hours (SEPH) (2011) [21], General Social Survey (GSS) (2010) [22], and Canadian Community Health Survey (CCHS) (2010) [23]. Some data were also drawn from scientific literature, as our secondary sources, as described in following sections. In the forthcoming paragraphs, we first describe the method for estimation of NPC and SNC incidence among workers exposed to wood dust and then we outline economic burden estimation methodology.

### 2.1. Incidence of Cancer among Exposed Workers

We drew the number of workers exposed to wood dust and their exposure level by occupational group, industrial sectors and provinces from CAREX Canada [15]. The incidence of occupational NPC and SNC attributable to wood dust were estimated by several members of our research team in the OCRC, using attributable fraction approach, similar to the one used in United Kingdom burden by Hutchings et al., [24] and Rushton et al., [25], but updated for Canadian context [14,26]. This methodology has been recently used to estimate the incidence of occupational lung cancers among Canadian workers exposed to diesel exhaust [27]. Calculating the attributable fraction involved three major steps. The first was to select an appropriate relative risk value from high-quality, epidemiological studies that are suitable for Canadian context. The second step was to assess the prevalence of exposure to wood dust among the Canadian working population. The exposure prevalence estimated based on the database previously developed by CAREX Canada [15]. The last step was population modelling. The number of workers ever exposed to wood dust during the risk exposure period (1961 to 2000) was calculated by counting the number of all workers exposed in the first year of the risk exposure period (1961) and number of new hires in each successive year (1962–2000). More details about this method were extensively described elsewhere [14,26].

### 2.2. Economic Burden Modelling

Economic burden of occupational cancers was categorized under three broad domains of direct costs (i.e., healthcare costs, out-of-pocket cost, and informal caregiving costs), indirect costs (i.e., labour productivity/output costs, employer adjustment costs, and home production losses), and intangible costs (i.e., health-related quality of life losses). All calculations were adjusted for age and sex and all monetary values were discounted to 2011 Canadian dollars, using a 3% discount rate. We also included a 1% (0.5–2% considered for sensitivity analysis) productivity growth rate for paid-labour market activity [18].

In order to estimate NPC and SNC healthcare costs, we classified costs under three distinct phases, namely: pre-diagnosis, treatment, and follow-up. To estimate pre-diagnosis and treatment costs, we drew on de Oliveira et al. who reported the healthcare costs of 21 common cancers occurring in Canada between 1997 and 2007 [28]. They captured several cost components including diagnostic tests, physician services, chemotherapy, radiotherapy, surgery, inpatient hospital admission, and outpatient drugs, visits, home care, continuing care, and long-term care. They placed all costs that happen 3 months before diagnosis under the pre-diagnosis phase, and all costs that happen in the subsequent 12 months after diagnosis under treatment costs. We drew on their study and considered CAD 1291 (CAD 1060–1522) as pre-diagnosis costs, for cases who survived beyond the first year after diagnosis, and CAD 2545 (CAD 2225–2865) for cases who died within the first year. We also considered CAD 28,775 (CAD 27,426–30,125) and CAD 48,214 (CAD 46,411–50,018) as treatment costs for cases survived beyond the first year after diagnosis and who died within the first year, respectively. Stage-specific survival probabilities of NPC and SNC were extracted from the American Joint Committee on Cancer [29] (Appendix A). We approximated the average recurrence rates of NPC at 17.7% [30] and SNC at 15.5% [31]. Different recurrence rates were investigated in sensitivity analyses, ranging between 12% and 22% [2]. We assumed CAD 1497 annual follow-up costs for survived cases for a 5-year period (Appendix B). For estimation of the follow-up costs, we drew on Ontario schedule facility fees and physician services under the health insurance act [17] and nasopharyngeal cancer treatment clinical practice guidelines [32].

Cancer cases encounter other costs associated with homemaking, complementary medicines, vitamins, supplements, travel, parking, accommodations and devices which are not usually reimbursed by the healthcare system. For an estimation of these out-of-pocket costs, we drew on a survey by Longo et al. [33]. They estimated the average monthly out-of-pocket costs of cancer treatment in Ontario at CAD 213 (CAD 0–704) with additional cost of CAD 372 (CAD 0–1066) related to travel costs. For survived cases, we considered out-of-pocket costs for a period of 3 years (1.5 years during treatment and 1.5 years for follow-up). However, this only captured a fraction of all out-of-pocket costs needed to deal with the cancers, as often these cancers have a follow-up treatment that will carry on for years. Regarding the informal caregiving costs of NPC and SNC cases, little has been published in the literature. We drew on Yabrof et al. and estimated the average time (months; hours per day) associated with informal caregiving for localized (12.2;7.8), regional (14.5;8.3), distance (17.9;10.9) and non-staged (12.8;9.8) cancer survivors separately [34]. We estimated the monetary values of informal caregiver’s time using worker’s minimum wages at CAD 9 (CAD 6–13) [35]. Then all survival, death, and recurrence probability data were combined with healthcare costs, out-of-pocket costs and informal caregiving costs through model that we developed in a spreadsheet.

There are different approaches to estimate labour productivity or output losses associated with morbidity and mortality. Human capital approach, which involves a societal perspective in assessing the forgone revenue and friction cost approach, which is a firm-level cost estimation method [36]. In this study, we used prior to estimate the losses of labour market productivity. Our estimates included lost wages and fringe benefits, multiplied by the time of absence from work due to sickness or premature mortality. An approach similar to friction cost was used to proxy the costs incurred by employers to replace absent workers [37]. In this approach, the monetary value of labour productivity or output lost due to morbidity (the friction period) was assumed to be 6 months of annual wage in the year of diagnosis. We used LFS and SLID to estimate average labour force wage [18,19], and then added 14% (10–20%) to account for payroll benefits, based on CSNA [20]. In order to estimate home production losses, we extracted the average time that individuals spend on various household-related activities from GSS [22] (Appendix C). These activities may include taking care of plants and animals, cooking, clean-up, auto maintenance, and other personal activities. For the estimation of the monetary value of this time, we used the minimum wage of housekeepers that we extracted from SEPH [21]. The productivity growth was assumed to be constant over time for home production and no payroll costs were considered.

NPC cases may encounter different kinds of dysfunctions associated with breathing, speech, vision, and hearing [38]. Similarly, SNC cases may feel symptoms such as nasal obstruction, facial pain, persistent rhinorrhea, and nosebleeds [39], which have a negative impact on their health-related quality of life. We compared the average health utility index (HUI) of cases with the general population, over 5 years after diagnosis for estimating their health-related quality of life losses. We approximated average HUI of stage I and II of all cases at 0.80 and stage III and IV at 0.74, based on Noel et al. [40], and the 5-year survival rate of cases extracted from Skarsgard et al. [41]. The general population’s average life expectancy and HUI were drawn from Canadian life tables [16] (Appendix D) and CCHS [23] (Appendix E). We used a conservative value of CAD 50,000 for an estimation of a Quality-Adjusted Life-Years (QALYs) monetary value [42]. Given the wide range of monetary values used for QALY in health economics literature, we considered a range of CAD 100,000/QALY [42] and CAD 150,000/QALY [43] for sensitivity analysis.

### 2.3. Sensitivity Analysis

Given the number of data elements required for the modeling and the variety of assumptions needed to proxy for the various cost components, the burden estimate is sensitive to the values used for key parameters. We investigated a variety of scenarios reflecting higher and lower bound values for healthcare costs, out-of-pocket costs, informal caregiving wage, recurrence rate, fringe benefit, productivity growth, monetary value per QALY, and cancers incidence.

## 3. Results

Table 1 represented incidence of NPC and SNC among workers exposed to wood dust in different provinces. From a total of 1,354,263 workers exposed to wood dust, 812,558 (60%) were men and 541,705 (40%) were women. From a total of 235 and 245 newly diagnosed cases of NPC and SNC in 2011, respectively, about 11 cases from each (95% Cl 9–29 and 95% Cl 6–32, respectively) were attributed to occupational exposure to wood dust. For both types of cancers, incidence was estimated to be higher for men than women. The estimated number of occupational cancers attributable to wood dust varies from the highest cases in British Columbia to the lowest in the province of Prince Edward Island. However, the data presented excludes the Yukon, Northwest Territories, and Nunavut, as there was not sufficient data to estimate cases for the territories. Occupational NPC and SNC in British Columbia (NPC: 3, SNC: 2), Ontario (NPC: 2, SNC: 3) and Quebec (NPC: 2, SNC: 3) have the highest incidence, due to their relatively large populations. The proportion of occupationally attributed cancers in British Columbia, Ontario and Quebec were 6.4%, 3.2% and 4.9% for NPC, and 10.7%, 6.5% and 9.7% for SNC, respectively. The lowest proportion of occupationally attributed cancer was estimated at less than one case in Prince Edward Island.

Table 2 presented the incidence of NPC and SNC among workers exposed to different level of wood dust by industrial sectors and occupational groups. Among workers exposed to wood dust, 379,228 (28%) were estimated to have low exposure and 575,834 (43%) and 399,201 (29%) were estimated to have medium and high exposure levels, respectively. Furthermore, the highest number of workers exposed to wood dust were in manufacturing (585,951), construction (560,139), and forestry and logging (51,251). Not surprisingly, the highest incidence of NPC and SNC were in the same industrial sectors. In terms of occupational group, the highest number of workers exposed to wood dust were in the construction trade (710,805), machine operators (215,202), and in processing, manufacturing, and utilities (101,095). Note that the incidence of NPC and SNC across industrial sectors and occupational groups vary for multiple reasons including the labour force size and exposure level.

Table 3 represents direct, indirect and intangible costs for NPC and SNC separately. For NPC, direct costs comprised approximately 20% of all costs while indirect costs and intangible costs constituted 55% and 25%, respectively, while for SNC the breakdown distribution was 16%, 42% and 42%, respectively. Direct costs were estimated to be CAD 1.08 million (CAD 98,500 per case) for NPC and CAD 1.07 million (CAD 99,214 per case) for SNC. For NPC, healthcare costs were approximately 11% of all costs while out-of-pocket cost and informal caregiving costs represented a marginal role at 4% and 5%. These fractions for SNC were 9%, 4% and 3%, respectively. Indirect costs for NPC and SNC were estimated at CAD 3.02 million (CAD 275,907 per case) and CAD 2.83 million (CAD 263,493 per case). For NPC, labour market productivity losses constituted approximately 48% of all costs, while home production losses as a result of morbidity and premature mortality and friction costs were 4% and 3%, respectively. These fractions for SNC were 36%, 3%, and 3%, respectively. We also estimated intangible costs at CAD 1.33 million (CAD 121,904 per case) for NPC and CAD 2.84 million (CAD 264,730 per case) for SNC, based on a value of CAD 50,000/QALY.

Table 4 provided the results of the one-way sensitivity analyses for each input variable. The estimated economic burden for NPC and SNC ranges from CAD 4.2- CAD 14.5 million and CAD 3.8- CAD 20.0 million. This translated to a −22% to 167% change for NPC and −43% to 196% for SNC. Sensitivity analyses indicated that incidence, monetary value of a QALY, out-of-pocket costs, and healthcare costs were parameters with the largest impact on the economic burden. One reason for the large range of NPC and SNC incidence was the wide range of attributable fractions that have been reported for these cancers in previously published literature.

## 4. Discussion

This study shed some light on prevalence of occupational exposure to wood dust amongst Canadian workers, incidence of NPC and SNC attributed to wood dust exposure, and their economic burden on society. From approximately 1.3 million workers exposed to wood dust, we expected 28%, 43% and 29% were exposed to low, medium, and high levels, respectively. We identified 4.6% of all NPC cases and 4.4% of all SNC cases attributed occupational exposure to wood dust. We estimated a total economic burden of occupational NPC and SNC of about CAD 12.2 million. Breakdown of our estimates between direct and indirect costs indicated 27% of NPC and SNC costs were associated with direct costs and 72% with indirect costs.

We considered conservative assumptions for the estimation of incidence and economic burden of NPC and SNC; thus, the real values were likely underestimated. Furthermore, our cases represented only a fraction of total occupational NPC and SNC cases in Canada, as there are other occupational agents such as formaldehyde, leather, and nickel, that are well-recognized for causing the same kind of cancers [11]. Regarding the incidence of NPC and SNC across occupations, not surprisingly, the highest cases were expected in manufacturing and construction, but surprisingly, a noticeable number of cancers can be expected in educational services, trade, and public administration occupational groups, to which less attention has been paid. It is worth noting that when generalizing these findings across countries, extra caution should be taken since many parameters such as availability of wood, types of technology being used, and working environment can make a considerable difference in exposure estimates [44].

To our knowledge, the present study is the first focused exclusively on the economic burden of occupational NPC and SNC, rather than population-level cancers. Consequently, it is difficult to compare our findings to other studies. The only comparable study in Canada is from Orenstein et al. who reported from six new cases of nasal sinuses cancer in Alberta, about two (ranges between one and three) cases were attributed to work [11]. However, their estimated incidence was not limited to occupational exposure to wood dust, and they also considered occupational exposure to other carcinogens such as formaldehyde, nickel, and mineral oils in their study. They estimated direct and indirect economic burden of nasal sinuses cancers at CAD 7977 and CAD 7129 per case in 2011 Canadian dollars. Their direct costs were lower than our estimate, as they only included healthcare costs, while out-of-pocket and informal caregiving costs were not considered in their estimates. Their indirect costs also were much lower than ours, as for productivity losses of mortality cases they only considered 18.1 days lost as an average. They also did not consider neither home production losses and employer’s friction costs nor cost related to losses of health-related quality of life in the cases. Breakdown of their reported values indicated that 53% of total costs were attributed to direct and 47% attributed to indirect costs.

Jacobson et al. reported the direct and indirect costs of cancers in the U.S. at CAD 95,848 and CAD 9629 per case in 2011 Canadian dollars, respectively [12]. In terms of direct cost, although they considered cost categories relating to hospital visits, office visits, emergency department, they did not consider out-of-pocket costs and informal caregiving costs. In terms of indirect costs, authors estimated the indirect costs through short-term disability of the cancer cases. They calculated monetary value of days lost using hourly wage CAD 29.37 in 2009 US dollars. But they did not consider the premature mortality and home production losses. The breakdown of their reported costs indicated that 91% of total cancer costs were attributed to direct and only 9% attributed to indirect costs.

Epstein et al. reported direct costs of oral and pharyngeal cancer cases at CAD 51,895 per case in 2011 Canadian dollars [13]. They considered healthcare costs based on the administrative claims data, but they did not estimate the out-of-pocket costs and informal caregiving costs of the patients. Additionally, they did not include indirect cost, or any value related to the quality of the life losses. It is important to be mentioned that when we compare our results with this study, we should be careful, not only because of the difference in cancers (i.e., oral and pharyngeal squamous cell carcinoma), but also because the sample was restricted to those who were continuously eligible for 1-year post-diagnosis. Therefore, this study has excluded patients who were diagnosed with and treated for the disease but died within the year.

The limitation of studies in the area of economic burden of occupational NPC and SNC may be attributed to two main reasons: a lack of methodological framework for estimation of the cancer cases attributed to occupational exposure; and lack of a framework for economic burden computation [10]. As a result, there is a great variety in terms of costs considered. Most considered only a narrow subset of the costs that comprise the societal burden. Some studies focused exclusively on healthcare costs and did not capture other costs such indirect or intangible costs. Such studies focused on the traditional insurance model, considering only provider costs, but the societal perspective. Although they provide useful information for insurers, they missed out a substantial portion of the societal burden and therefore may lead to suboptimal policy decision making.

The key strengths of our study were the detailed approach to estimate the economic burden of cancer cases under three categories—direct, indirect and intangible—which is more comprehensive in the terms of costs considered than most of the previous economic burden studies about these cancers. Furthermore, the model contains a large amount of detailed information on healthcare costs, personal earning losses and intangible losses that mainly incur to workers and their families. We incorporated several Canadian data sources to account for sex, age, province, occupational group, and industrial sectors in our model. We developed a framework based on several previously published papers to estimate the stage distribution, survival and recurrence rate of different stages of cancers, which allowed us to have better picture of the actual economic burden of occupational NPC and SNC related to wood dust in whole paradigm of disease. Our study took a lifetime case costing approach, considering factors such as diagnosis, survival probabilities, recurrence probabilities, and death rate. We captured a substantial portion of what the occupational health and safety literature describes as the hidden part of the cost’s iceberg [45]. Additionally, in this study, we used incidence costing approach, which is preferred to the prevalence costing approach, from both occupational health and safety professionals and policy decision-maker perspectives. This approach fits well with investment decision making which requires estimates of future costs. Thus, our study provides not only estimates of the economic burden of NPC and SNC, but also can serve as an example for future economic burden or cost of illness studies.

Lack of data for key input parameters has often been cited as a limitation in occupational disease burden studies, which also was the case with our study. Some assumptions (and sometimes compromises) were made to address data gaps. We considered conservative assumptions; thus, the real economic burden of occupational NPC and SNC was likely underestimated. Regarding the informal caregiving costs, we only considered a fraction of the real costs based on the time that they spend; however, sometimes informal caregivers in practice may encounter challenges for finding a flexible job and may deal with significant losses in terms of paid work. Another limitation was related to the estimation of productivity losses as NPC and SNC may lead to other forms of work productivity losses such as presenteeism (i.e., reduced productivity while at work), reduced team effectiveness, and penalties associated with late production [46]. However, we only included labour productivity losses based on the human capital approach. Although, there was some uncertainty associated with some input parameters, the sensitivity analysis indicated how changing input data can affect our estimated economic burden under different scenarios.

## 5. Conclusions

The present study captured a significant portion of the burden of newly diagnosed occupational NPC and SNC amongst workers exposed to wood dust in Canada. The findings of this study provide invaluable information for policymakers to promote prevention strategies in order to enhance the current and future health of workers in industrial sectors where exposure to wood dust occurs. Furthermore, the estimated per-case costs of this study, under three categories—direct, indirect and intangible costs—can serve as inputs for the economic evaluation of occupational health and safety intervention (i.e., cost-benefit/cost-effectiveness) to demonstrate the monetary impact of decreasing or eliminating occupational exposure by averted NPC and SNC cases. Our estimates also can raise awareness about the risk of occupational exposure to wood dust, especially in countries with a large wood-related industry.

## Figures and Tables

**Table 1 ijerph-19-01144-t001:** Incidence of nasopharynx cancer (NPC) and sinonasal cancer (SNC) among workers exposed to wood dust in Canada.

	Exposed workers ^a^	Total NPC ^b^	Occupational NPC ^c^	Total SNC ^d^	Occupational SNC ^e^
Total					
	1,354,263	235	11 (4.6%)	245	11 (4.4%)
Sex					
Men	812,558 (60%)	165	11 (6.4%)	145	10 (7.0%)
Women	541,705 (40%)	70	0 (0.4%)	100	1 (0.5%)
Provinces of residence					
AB	61,553 (9.1%)	20	1 (4.0%)	3	1 (26.7%)
BC	141,639 (21.0%)	45	3 (6.4%)	15	2 (10.7%)
MB	680,186 (3.5%)	5	0 (6.0%)	5	0 (8.0%)
NB	23,550 (4.1%)	5	0 (0.4%)	5	1 (14.0%)
NL	27,374 (2.4%)	10	0 (0.2%)	3	0 (16.7%)
NS	16,035 (3.5%)	10	1 (9.0%)	5	1 (10.0%)
ON	23,753 (27.7%)	75	2 (3.2%)	48	3 (6.5%)
PE	186,830 (0.5%)	0	0 (0%)	0	0 (0%)
QC	3465 (25.4%)	35	2 (4.9%)	33	3 (9.7%)
SK	171,352 (2.7%)	0	0 (0%)	5	0 (6.0%)

Note: ^a^ total number of the workers exposed to wood dust in 2011, ^b^ incidence of nasopharynx cancer, ^c^ nasopharynx cancer cases attributed to occupational wood dust exposure, ^d^ incidence of sinonasal cancer, ^e^ sinonasal cancer cases attributed to occupational wood dust exposure. Canadian provinces are: AB—Alberta; BC—British Columbia; MB—Manitoba; NB—New Brunswick; NL—Newfoundland and Labrador; NS—Nova Scotia; ON—Ontario; PE—Prince Edward Island; QC—Quebec; SK—Saskatchewan. Owing to rounding, columns and rows may not sum to the exact same value.

**Table 2 ijerph-19-01144-t002:** Incidence of nasopharynx cancer (NPC) and sinonasal cancer (SNC) among workers exposed to different levels of wood dust by industry and occupation.

	Exposed workers ^a^	Low (%)	Medium (%)	High (%)	NPC Cases (%) ^b^	SNC Cases (%) ^c^
Total						
	100 1,354,263	28 (379,228)	43 (575,834)	29 (399,201)	100 (11)	100 (11)
Industrial sector						
Accommodation and food services	7131	100	0	0	0 (<0.01)	0 (0.5)
Agriculture	2681	97	3	0	0 (0.01)	0 (0.2)
Business/management/other support	2229	100	0	0	0 (<0.01)	0 (0.2)
Construction	560,139	27	73	0	5 (43.4)	4 (38.5)
Educational services	23,491	31	4	65	0 (1.1)	0 (1.7)
Finance/insurance/real estate/leasing	7638	100	0	0	0 (<0.01)	0 (0.5)
Fishing/hunting/trap	81	100	0	0	0 (<0.01)	0 (0.01)
Forestry and logging	51,251	47	1	52	0 (3.0)	0 (3.9)
Health care/social assistance	7592	100	0	0	0 (<0.01)	0 (0.5)
Info/culture/recreation	8224	100	0	0	0 (<0.01)	0 (0.5)
Manufacturing	585,951	12	27	61	6 (51.7)	5 (47.2)
Mining/oil/gas extract	5046	100	0	0	0 (<0.01)	0 (0.4)
Other services	9469	100	0	0	0 (<0.01)	0 (0.6)
Professional scientific/technical service	7896	97	3	0	0 (0.02)	0 (0.5)
Public administration	19,832	84	16	0	0 (0.4)	0 (1.4)
Trade	44,517	92	8	0	0 (0.4)	0 (2.7)
Transportation/warehousing	7405	100	0	0	0 (<0.01)	0 (0.5)
Utilities	3689	100	0	0	0 (<0.01)	0 (0.3)
Occupational group						
Construction trades	710,805	26	60	14	6 (54.1)	5 (50.9)
Contractors, supervisors in trades, transportation	94,732	100	0	0	0 (0.01)	1(6.5)
Labourer in processing, manufacturing, utilities	101,095	6	11	84	1 (9.7)	1(7.9)
Machine operators and assemblers in manufacturing	215,202	9	19	72	2 (20.2)	2 (18.4)
Natural and applied sciences	3438	100	0	0	0 (<0.01)	0 (0.2)
Art, culture, recreation and sport	315	19	81	0	0 (0.02)	0 (4.3)
Social science/government service/religion	76	100	0	0	0 (<0.01)	0 (0.02)
Occupations unique to primary industry	57,281	52	1	47	0 (3.0)	0 (0.01)
Other management occupations	310	100	0	0%	0 (<0.01)	0 (0.02)
Other trades occupations	75,231	20	80	0	1 (6)	1 (4.9)
Sales and service occupations	9132	4	96	0	0 (0.9)	0 (0.6)
Teachers and professors	16,185	0	6	94	0 (1.1)	0 (1.2)
Trades helpers, construction, transportation labourers	69,002	33	39	29	1 (4.9)	1 (5.1)
Transport and equipment operators	1459	75	25	0	0 (0.04)	0 (0.1)

Note. ^a^ total number of the workers exposed to wood dust in 2011, ^b^ incidence of occupational nasopharynx cancer, ^c^ incidence of occupational sinonasal cancer. Owing to rounding, columns and rows may not sum to the exact same value.

**Table 3 ijerph-19-01144-t003:** Total economic burden of occupational nasopharynx cancer (NPC) and sinonasal cancer (SNC).

Type of Cancer	NPC			SNC		
	Per-Case	Total	%	Per-Case	Total	%
Direct costs						
Healthcare cost	CAD 52,531	CAD 575,206	11	CAD 53,275	CAD 572,459	9
Out-of-pocket cost	CAD 21,060	CAD 230,603	4	CAD 21,060	CAD 226,296	3
Informal caregiving	CAD 24,909	CAD 272,749	5	CAD 24,879	CAD 267,328	4
Sum	CAD 98,500	CAD 1,078,558	20	CAD 99,214	CAD 1,066,083	16
Indirect costs						
Productivity losses	CAD 240,201	CAD 2,630,149	48	CAD 228,334	CAD 2,453,514	36
Home production losses	CAD 18,873	CAD 206,655	4	CAD 18,895	CAD 203,035	3
Friction losses	CAD 16,833	CAD 184,322	3	CAD 16,264	CAD 174,767	3
Sum	CAD 275,907	CAD 3,021,126	55	CAD 263,493	CAD 2,831,316	42
Intangible costs						
Health-related quality of life losses (QALY)	2.4	26.7	-	5.3	56.9	-
Monetary value of health-related quality of life losses (CAD 50 k/QALY)	CAD 121,904	CAD 1,334,824	25	CAD 264,730	CAD 2,844,602	42
Sum	CAD 496,311	CAD 5,434,508	100	CAD 627,437	CAD 6,742,000	100

Notes. Owing to rounding, columns, and rows may not sum to the exact same value, All values are in 2011 Canadian dollars.

**Table 4 ijerph-19-01144-t004:** Analysis of sensitivity of cost categories to different scenarios for nasopharynx cancer (NPC) and sinonasal cancer (SNC).

Assumptions	Range	NPC ^a^	Change	SNC ^b^	Change
Healthcare costs (pre-diagnosis; initial phases)	CAD 32,659; CAD 55,761	CAD 5.28 M	−2.9%	CAD 6.72 M	−0.4%
	CAD 36,283; CAD 60,630	CAD 5.46 M	0.5%	CAD 6.77 M	0.4%
Out-of-pocket costs (per month)	CAD 400	CAD 5.20 M	−4.2%	CAD 6.52 M	−3.4%
	CAD 600	CAD 5.90 M	8.6%	CAD 7.20 M	6.8%
Informal caregiving wage (per hours)	CAD 6	CAD 5.34 M	−1.8%	CAD 6.65 M	−1.4%
	CAD 13	CAD 5.54 M	1.9%	CAD 6.84 M	1.5%
Recurrence rate	12%	CAD 5.41 M	−0.5%	CAD 6.65 M	−1.4%
	22%	CAD 5.46 M	0.5%	CAD 6.84 M	1.5%
Fringe benefit	10%	CAD 5.34 M	−1.8%	CAD 6.65 M	−1.4%
	20%	CAD 5.58 M	2.7%	CAD 6.88 M	2.1%
Productivity growth	0.5%	CAD 5.38 M	−1.0%	CAD 6.69 M	−0.8%
	2%	CAD 5.55 M	2.1%	CAD 6.85 M	1.6%
Monetary value per QALY ^c^	CAD 100,000	CAD 6.77 M	24.6%	CAD 8.14 M	20.7%
	CAD 150,000	CAD 8.10 M	49.1%	CAD 10.26 M	52.1%
Cases (Cl of 95%) ^d^	NPC:9; SNC:6	CAD 4.22 M	−22.4%	CAD 3.82 M	−43.3%
	NPC:29; SNC:32	CAD 14.51 M	167.0%	CAD 19.98 M	196.3%

Notes. ^a^ Total economic burden of nasopharynx cancer, ^b^ Total economic burden of sinonasal cancer, ^c^ assumeing different monetary values for a Quality-Adjusted Life-Years. ^d^ lower and higher bound of the expected cancer cases with confidence intervenal of 95%. Owing to rounding columns and rows may not sum. All monetary values are in 2011 Canadian dollars and M denotes that the units of figures presented are in millions.

## Data Availability

The data presented in this study are available within the article and in supplementary material.

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
