# Peer review of "Occupational Exposure to Wood Dust and the Burden of Nasopharynx and Sinonasal Cancer in Canada"

_ijerph, 2022, doi:10.3390/ijerph19031144_

Round 1

Reviewer 1 Report

The authors report a study entitled "Occupational Exposure to Wood Dust and the Burden of Nasopharynx and Sinonasal Cancer in Canada". The article is very interesting, well written and the analysis correct. The results support the stated objective and the conclusions are consistent. I suggest publication in present form.

Author Response

Authors Response: Thanks for your time to review this paper.

Reviewer 2 Report

Estimated Authors,

Estimated Editors of IJERPH,

I've read with very great interest the present paper from the study group lead by DEMERS, with a well-known expertise on wood-dust related disorders. 

In this study, MOFIDI et al have performed an estimation on the incidence and economic burden of occupational NPC and SNC, based on Canadian data. In summary:

1.3 million workers were exposed to various levels of wood dusts,;

among the the 235 new diagnoses of NPC and 245 new diagnoses of SNC, only 11 cases for both groups found a clear occupational exposure, with a resulting burden of around 5.4 and 6.7 million canadian dollar per case. 

The study is well performed and accurately documented. From a methodological point of view, I've no recommendations or requirements. I only recommend the Authors to discuss more extensively the burden of work-related cases of nasopharyngeal and sinonasal cancers, as the raw numbers of cases (11 + 11) represent very small share (4.6% + 4.4%) of new diagnoses, and previous reports (e.g. https://oem.bmj.com/content/57/6/376) accounted for a very higher esimate (16%). Some explanations may therefore significantly improve the quality of this othewise well written paper.

Author Response

Authors Response: Thanks for your time to review this paper and thanks for your insightful comments. We reviewed Vaughan et al., (2000) study and noted that “16%” is related to formaldehyde exposed workers, not wood dust exposed workers. We brought part of the text here: “If occupational exposure to formaldehyde is a causal factor, we estimate that such exposure accounts for about 16% of squamous cell and unspecified carcinomas of the nasopharynx in the United States.” For wood exposed workers Vaughan et al., (2000) reported as follows: “among the 196 cases of carcinoma of any histological type, 22 (11.2%) had worked in jobs judged to entail exposure to wood dust, compared with 24 (9.8%) controls”. So, the crude OR of 1.2 increased to 1.3 (95% CI 0.6 to 2.6) after controlling for age, sex, race, education, SEER site, proxy status, and cigarette use. Additional control for cumulative exposure to formaldehyde reduced the OR to 1.1 (95% CI 0.5 to 2.3).” Although we aware your concern regarding the difference across studies and to address that under the discussion section we elaborated as follow: “It is worth noting that when generalizing these findings across countries extra caution should be taken since many parameters such as availability of wood, types of technology being used, and working environment can make a considerable difference in exposure estimates (Olsson et al., 2021).Also, in several sections of the paper, we point out to the underestimation as follow: “We considered conservative assumptions for estimation of incidence and economic burden of NPC and SNC, thus, the real values were likely underestimated.”

Reviewer 3 Report

In this study, the authors wanted to calculate the cost of wood dust tumours. The objective is interesting and there are not many studies of this type. Originality is an advantage of this study, which reduces the possibility of comparing the results with those of other studies. Some minor remarks may improve the manuscript. Please note that it is not mandatory that the authors cite the specific articles, and they are welcome to seek alternative manuscripts in the literature that are relevant to the manuscript’s content.

  1. The authors did not follow the editorial rules for the indication of references. They didn't use the template and didn't number the lines; this hinders the indication of possible corrections.
  2. Introduction, 2nd Among the wood dust exposures, authors should add carpentry activities [Bevilacqua L, Sacco A, Magnavita N. Health surveillance audit of wood dust exposure. Med Lav 2003; 94: 224-30].
  3. The authors should consider that the composition and intensity of process-generated substances can vary substantially, depending on the parameters of the underlying processes [Olsson A, Kromhout H. Occupational cancer burden: the contribution of exposure to process-generated substances at the workplace. Mol Oncol. 2021 Mar;15(3):753-763. doi: 10.1002/1878-0261.12925].

Author Response

Authors Response: Thanks for your time to review this paper and thanks for your insightful comments. Please note following modifications, in response to your comments:

  1. In the final version of the paper, formatting issue has been addressed.
  2. In the modified version of the paper, we added “carpentry” activated under the introduction, 2nd paragraph for clarification. Although since we could not access the full version of Bevilacqua et al., (2003) study in English language (since the article has been published in Italian), we decided to do not add it in our citation list.
  3. We agree that wood dust exposure is more prevalent in countries with larger wood-related industries, as we pointed out to this fact in our introduction. Certainly, interpretation and generalization of these findings for other contexts need extra caution. Therefore, responding to your comment, we elaborated on our discussion, 3rd paragraph, as follow: “It is worth noting that when generalizing these findings across countries extra caution should be taken since many parameters such as availability of wood, types of technology being used, and working environment can make a considerable difference in exposure estimates (Olsson et al., 2021).”